# Toward Oral Supplementation of Diamine Oxidase for the Treatment of Histamine Intolerance

**DOI:** 10.3390/nu14132621

**Published:** 2022-06-24

**Authors:** Lucas Kettner, Ines Seitl, Lutz Fischer

**Affiliations:** Department of Biotechnology and Enzyme Science, Institute of Food Science and Biotechnology, University of Hohenheim, Garbenstr. 25, 70599 Stuttgart, Germany; lucas.kettner@uni-hohenheim.de (L.K.); ines.seitl@uni-hohenheim.de (I.S.)

**Keywords:** diamine oxidase, histamine, dietary supplement, *Yarrowia lipolytica*, histamine intolerance, biogenic amines

## Abstract

A new diamine oxidase (DAO-1) was discovered recently in the yeast *Yarrowia lipolytica* PO1f and investigated for its histamine degradation capability under simulated intestinal conditions. DAO-1 was formulated together with catalase as a sucrose-based tablet. The latter (9 × 7 mm; 400 mg) contained 690 nkat of DAO-1 activity, which was obtained from a bioreactor cultivation of a genetically modified *Y. lipolytica* with optimized downstream processing. The DAO-1 tablet was tested in a histamine bioconversion experiment under simulated intestinal conditions in the presence of food constituents, whereby about 30% of the histamine was degraded in 90 min. This amount might already be sufficient to help people with histamine intolerance. Furthermore, it was found that the stability of DAO-1 in a simulated intestinal fluid is influenced distinctively by the presence of a food matrix, indicating that the amount and type of food consumed affect the oral supplementation with DAO. This study showed for the first time that a microbial DAO could have the potential for the treatment of histamine intolerance by oral supplementation.

## 1. Introduction

The histamine, one of the biogenic amines, is associated with increasing numbers of foodborne illness outbreaks in the European Union as stated by the European Food and Safety Authority and the European Centre for Disease Prevention and Control [1]. Thereby, foods containing histamine levels of 500 mg·kg^−1^ and above could be considered as hazardous to human health [2]. However, the consumption of foods with moderate or even low histamine concentrations also negatively affects humans who suffer from a histamine intolerance [3]. It is discussed that around 1% of the total population might be affected [4]. Typical symptoms of histamine intolerance are gastrointestinal disorders, headaches, asthma, flushing, and sneezing [3]. The intolerance results from an imbalance between the uptake of histamine and the histamine-degrading enzyme diamine oxidase (DAO) (EC 1.4.3.22) [5]. Human DAO is a secretory enzyme that is located mainly in the small intestinal mucosa and the kidneys [3,6,7]. This enzyme catalyzes the oxidative deamination of histamine or other biogenic amines to the corresponding aldehydes, ammonia, and hydrogen peroxide [8]. Several factors affect the available activity of DAO in humans. Different single-nucleotide polymorphisms have been associated with lower transcriptional activity of the DAO gene or with a reduction in the DAO enzyme functionality [9]. The serum DAO activity was found to be significantly lower in patients suffering from symptoms of histamine intolerance, suggesting that it might be used as a diagnostic tool [5,10]. Additionally, it was shown that the serum DAO activity seemed to be in direct correlation with the health status of the intestinal mucosa [11]. A reduced activity of DAO can, therefore, be observed for various gastrointestinal disorders and injuries [9]. However, this can be only a temporary effect as it was shown in patients undergoing chemotherapy that they were able to recover from decreased serum DAO activities within a few weeks [12]. Furthermore, the DAO activity available is affected by the intake of other biogenic amines, drugs, or alcohol [3]. An efficient degradation by DAO on-site is of high relevance because dietary histamine enters the body primarily through the small intestine. Therefore, a solution approach could be the supplementation of exogenous DAO to support the insufficient human DAO. Commercially available dietary supplements that contain DAO from a pig kidney extract have been investigated for the potential in the treatment of histamine intolerance in several clinical studies [13,14,15,16,17]. Thereby, it was found to be effective in the treatment of histamine intolerance-associated symptoms. However, this kind of preparation was biochemically investigated recently, and no DAO activity was determined [18]. It was generally proven that high DAO activities of at least 50 nkat are needed to degrade food-relevant histamine amounts [18]. The extraction of DAO from animal-based sources, such as pig kidney, seems to be inefficient to provide the necessary level of activity. The superior approach is the overexpression of DAO in a suitable microbial host as it has already been shown in multiple cases, such as in the microbial production of calf chymosin for cheese making [19]. The microbial overexpression would provide sufficient DAO activities for the preparation of highly efficient DAO tablets for oral supplementation. A new DAO-1 was discovered recently in *Y. lipolytica* and biochemically characterized [20]. This DAO-1 showed promising characteristics for administration in the food industry or as a dietary supplement as it was able to efficiently degrade not only histamine but also other food-relevant biogenic amines, such as tyramine, putrescine, and cadaverine.

The aim of this study was to investigate the potential of the newly discovered DAO-1 from *Y. lipolytica* as an oral supplement for the treatment of histamine intolerance. Accordingly, DAO-1 was formulated as a tablet and applied for the conversion of high histamine amounts under simulated intestinal conditions. Furthermore, the stability and kinetics of DAO-1 under these conditions were assessed.

## 2. Materials and Methods

### 2.1. Materials and Reagents

1,4-piperazinediethanesulfonic acid (PIPES), histamine dihydrochloride, sodium hydroxide (NaOH), D(+)-sucrose, monobasic potassium phosphate (KH_2_PO_4_), hydrochloric acid (HCl), and hydrogen peroxide (30%) were purchased from Carl Roth GmbH (Karlsruhe, Germany). Sodium dihydrogen phosphate, sodium diethyldithiocarbamate, ortho-phosphoric acid (H_3_PO_4_), and thiamine chloride dihydrochloride were purchased from Merck (Darmstadt, Germany). Bovine serum albumin (BSA; modified Cohn Fraction V, pH 5.2) was purchased from Serva electrophoresis GmbH (Heidelberg, Germany). Catalase (from *Micrococcus lysodeikticus*; 111,700 U·mL^−1^) and pancreatin from porcine pancreas (8× USP specifications) were purchased from Sigma-Aldrich (Merck) (St. Louis, MO, USA). (10-(carboxymethyl-aminocarbonyl)-3,7-bis(dimethylamino) phenothiazine sodium salt (DA-67) was purchased from Fujifilm Wako Chemicals U.S.A. Corp (Richmond, VA, USA). Horseradish peroxidase (Grade I) was purchased from AppliChem GmbH (Darmstadt, Germany). Whey protein isolate (WPI; 90% (*w*/*w*) protein) was obtained from Sachsenmilch Leppersdorf GmbH (Leppersdorf, Germany). Sodium caseinate (90.6% (*w*/*w*) protein) was obtained from FrieslandCampina (Amersfoort, Netherlands).

### 2.2. Production and Purification of DAO-1

DAO-1 (Uniprot: DAO-1 (Q6CGT2)) was produced using a genetically modified *Y. lipolytica* PO1f strain (*Y. lipolytica* PO1f_*axp*_*dao-1*), which was obtained from the work of Kettner et al. [20]. Here, the native DAO-1 gene was integrated into the *axp* locus on the genome of *Y. lipolytica* using the CRISPR-cas9 system. The DAO-1 expression was conducted using the strong and constitutive UAS1B8-TEF(136) promotor. *Y. lipolytica* PO1f_*axp*_*dao-1* was cultivated in the Labfors 5 bioreactor system (Infors GmbH, Einsbach Germany) in a working volume of 5 L, according to Kettner et al. [20]. Cells were harvested after 56 h of cultivation and were stored at −20 °C until they were disrupted. For disruption, 150 g of cells was thawed on ice and used to prepare a 20% (*w*/*w*) suspension in PIPES buffer (25 mM, pH 7). The cells were disrupted in a bead mill (Dyno^®^-Mill Typ KDL A; Willy A. Bachofen AG Maschinenfabrik, Muttenz, Swiss) at 2500 rpm using glass beads with a diameter of 0.75 mm. The system was cooled to 5 °C using an Ultra-Kryomat^®^ RUK50 (Lauda Dr. R. Wobser GmbH & Co. KG, Lauda-Königshofen, Germany). The cell suspension was fed to the Dyno^®^-Mill system continuously with a peristaltic pump at a rate of around 14 mL·min^−1^, providing a residence time of 18 min. Afterwards, the glass beads were washed with 750 mL PIPES buffer (25 mM, pH 7) at around 14 mL·min^−1^ with the bead mill still running at 2500 rpm. The initial cell lysate and the buffer used for washing the glass beads were pooled and centrifuged (8000× *g*, 4 °C, 10 min). Around 800 mL of supernatant were collected and further purified by ammonium sulfate precipitation (60% (*v*/*v*) 4 M (NH_4_)_2_SO_4_) and hydrophobic interaction chromatography, according to Kettner et al. [20]. The purified DAO-1 was stored at −80 °C.

### 2.3. DAO-1 Activity Determination

DAO-1 activity was determined using the colorimetric DA-67 enzyme assay [21]. The reaction mixture, containing 375 µL histamine solution (30 mM; dissolved in 25 mM PIPES; pH 7.2) and 363 µL DA-67 reagent (10-(carboxymethyl-aminocarbonyl)-3,7-bis(dimethylamino) phenothiazine sodium salt; 50 µM; dissolved in 25 mM PIPES; pH 7.2), was incubated at 37 °C for 10 min and stirred at 750 rpm. Subsequently, 12 µL (266 units·mL^−1^) of horseradish peroxidase (Grade I) was added. The reaction was started by the addition of 25 µL DAO solution and incubated at 37 °C and 750 rpm. The reaction was stopped by the addition of 50 µL sodium diethyldithiocarbamate (30 mM). After centrifugation (10,000× *g*, 3 min, 20 °C), the absorption was measured at 620 nm. The histamine solution was replaced with buffer (25 mM PIPES; pH 7.2) for reference. Hydrogen peroxide (0.5–10 nmol·mL^−1^) was used for the calibration. The enzyme activity was calculated in nkat, whereby 1 nkat converts 1 nmol substrate·s^−1^ at 37 °C.

### 2.4. Protein Analysis

The protein content of the DAO-1 preparation was determined according to Bradford, using BSA as a standard [22]. Additionally, the DAO-1 preparation used for the preparation of the tablets was investigated by sodium dodecyl sulfate-polyacrylamide gel electrophoresis (SDS-PAGE) on a 10% separating gel [23]. An amount of 5 µg of protein was loaded onto the SDS-PAGE. A protein molecular mass standard was used (Precision Plus Protein™ unstained protein standard 10–250 kDa) for molecular mass determination. Coomassie Brilliant Blue G-250 was used to stain the gel [24].

### 2.5. Stability of DAO-1 in Simulated Intestinal Fluid (SIF)

The stability of DAO-1 was tested in an SIF with and without the addition of established food constituents to simulate possible food matrices. Therefore, pancreatin-containing SIF was prepared, as described in the United States Pharmacopeia [25]. Different food matrix stock solutions (4× concentrated) were prepared as follows: 200 g BSA·L^−1^ and 100 g sucrose·L^−1^ (=food matrix 1), 100 g BSA·L^−1^, 100 g WPI·L^−1^, and 200 g sucrose·L^−1^ (=food matrix 2), and BSA, WPI, and sodium caseinate each at 66.68 g·L^−1^ and 200 g sucrose·L^−1^ (=food matrix 3). The pH of each stock solution was adjusted to 6.8 with 1 M NaOH. The freshly prepared 2× concentrated SIF, the food matrix stock solutions, and DAO-1 (desalted against H_2_O_dd_ using PD MidiTrap G-25 columns; GE Healthcare, Chicago, IL, USA) were individually incubated for 5 min in a thermoshaker at 37 °C. Subsequently, 500 µL of 2× concentrated SIF, 250 µL of DAO-1, and 250 µL of the food matrix stock solutions were combined and incubated at 37 °C and 500 rpm in a thermoshaker. In order to test the stability of DAO-1 in the absence of food constituents, H_2_O_dd_ was added instead of the food matrix stock solution. A sample of 100 µL was taken immediately after combining the solutions and used for DAO-1 activity determination using the DA-67 assay.

### 2.6. Kinetics of DAO-1 in an SIF

The apparent kinetic parameters of DAO-1 were determined by Michaelis–Menten kinetics with histamine as the substrate (1.56 to 50 mM) in a pancreatin-free and -treated food matrix SIF 3. DAO-1 activity was determined using the DA-67 assay. Regarding the pancreatin-free approach, the DA-67 reagent (50 µM) was prepared in pancreatin-free SIF containing BSA, WPI, and sodium caseinate each at 35.6 g·L^−1^ and 106.8 g sucrose·L^−1^ (pH 6.8). The histamine and horseradish peroxidase were dissolved in pancreatin-free SIF. The pH for histamine was readjusted to 6.8 using 1M NaOH. DAO-1 was desalted against pancreatin-free SIF using PD MidiTrap G-25 columns.

Concerning the pancreatin-treated approach, BSA, WPI, and sodium caseinate each at 16.67 g·L^−1^ and 50 g sucrose·L^−1^ were first incubated in SIF (with pancreatin) at 37 °C and 130 rpm for 90 min. The hydrolysis was stopped by heating the solution at 95 °C for 15 min. Subsequently, it was centrifuged (8000× *g*, 4 °C, 10 min). The supernatant was then used to dissolve the DA-67 reagent (50 µM), histamine, and horseradish peroxidase, and to dilute DAO-1. The pH of the histamine solution was readjusted to 6.8 using 1 M NaOH. Histamine was replaced by hydrogen peroxide (0.5–20 nmoL·mL^−1^) for the calibration. Kinetic investigations were done within the initial reaction velocity.

### 2.7. Preparation of DAO-1 Tablets

The purified DAO-1 was first concentrated by ammonium sulfate precipitation. Therefore, liquid ammonium sulfate (4 M) was added dropwise under stirring and on ice to 335 mL purified DAO-1 solution to a final concentration of 60% (*v*/*v*). After completing the addition of liquid ammonium sulfate, the approach was further incubated for 60 min on ice. It was then centrifuged (8000× *g*, 4 °C, 25 min). The supernatant was completely removed and the pellet was dissolved in 3 mL sodium phosphate buffer (20 mM, pH 7). Sucrose powder was added to a final concentration of 40 g·L^−1^. Furthermore, 60 µkat of catalase from *M. lysodeikticus* was added. The final DAO-1 solution was divided into four parts which were separated in weighed 2 mL Eppendorf reaction tubes. These were frozen at −80 °C before they were freeze-dried. The freeze-dried powders were mixed with sucrose at a ratio of 50/50 (*w*/*w*) before they were used to prepare the DAO-1 tablets with a self-built tablet press (Appendix A).

### 2.8. Histamine Bioconversion Using DAO-1 Tablets

A histamine bioconversion was done using the DAO-1 tablets in a food-relevant histamine concentration of 1.35 mM (150 mg·L^−1^), as described in Kettner et al. [20]. The experiment was performed in a 500 mL approach volume in 1 L Erlenmeyer flasks that contained the food matrix SIF 3 (BSA, WPI, and sodium caseinate at 16.67 g·L^−1^ and sucrose at 50 g·L^−1^ in SIF (pH 6.8)) and 75 mg of histamine. The bioconversion was done in triplicate. The approaches were preincubated at 37 °C for 2 h. Pancreatin (8× USP specifications) (20 g·L^−1^ in SIF) was preincubated for 5 min at 37 °C and added to a final concentration of 1.25 g·L^−1^. Immediately after mixing the approaches, samples of 2 mL were taken, which were inactivated at 95 °C for 5 min in a water bath and then treated as described in Section 2.9. Furthermore, a 20 mL sample was taken from a reference bioconversion approach (without histamine), which was cooled down in an ice-water bath for the subsequent preparation of a histamine calibration for the reversed-phase high-performance liquid chromatography (RP-HPLC) analysis of the initial histamine concentration. Accordingly, a histamine stock solution was diluted in the reference approach media to histamine concentrations between 0.25 and 2 mM. These calibration samples were heated at 95 °C for 5 min in a water bath and then treated as described in Section 2.9. The histamine bioconversion was started by the addition of one DAO-1 tabletto each approach. Additionally, a DAO-1 tablet was also added to the reference approach. No DAO-1 tablet was added to the negative control approach. The flasks were incubated on a rotary shaker at 37 °C and 130 rpm for 90 min. Samples of 2 mL were taken after 30, 50, 70, and 90 min, inactivated at 95 °C for 5 min in a water bath and treated as described in Section 2.9. After 90 min, a sample of 20 mL was taken from the reference approach and cooled down in an ice-water bath. Subsequently, histamine calibration samples (0.1–1.5 mM) for the RP-HPLC analysis of the histamine concentration in the bioconversion samples (30–90 min) were prepared as described above.

### 2.9. Sample Preparation of Bioconversion Samples for the RP-HPLC Analysis

Heat-inactivated samples from the histamine bioconversion were cooled down on ice before they were centrifuged (10,000× *g*, 4 °C, 3 min). The supernatant (1 mL) was loaded on a PD MidiTrap G-25 column which was equilibrated with H_2_O_dd_. Undigested proteins and large peptides were eluted from the column using 1.5 mL H_2_O_dd_ and discarded. Histamine and molecules of low molecular weight were eluted in 2 mL H_2_O_dd_ and collected. The pH value of these samples was adjusted to around 2 using 35 µL HCl (1 M). The samples were kept at 20 °C in a thermoshaker before they were further purified.

The cation exchange material Lewatit^®^S100 (275 mg) was filled into a 1 mL pipette tip, which was loosely sealed at the bottom and top with cotton wool. The material was then washed with 4 mL of H_2_O_dd_. Afterwards, it was equilibrated with 4 mL of HCl (10 mM). The pH-adjusted bioconversion samples were then applied (1 mL each) to the cation exchange material. The material was washed with 5 mL of H_2_O_dd_. A volume of 600 µL of ammonia (4 M) was then added to the material and discarded. Again, 600 µL of ammonia (4 M) was added to elute the histamine. The ammonia water was evaporated at 70 °C and 500 rpm in a thermoshaker overnight. The remains were dissolved in 200 µL HCl (10 mM). A volume of 50 µL of internal standard solution (thiamine chloride dihydrochloride; 6 mM in H_2_O_dd_) was then added for the RP-HPLC analysis. The pH of each sample was adjusted to around 2 by the addition of 5 µL HCl (1 M). Samples were centrifuged (20,000× *g* 4 °C, 5 min) before they were analyzed by RP-HPLC.

### 2.10. RP-HPLC Analysis of Histamine in Bioconversion Samples

The histamine concentration in the bioconversion samples was determined by RP-HPLC, according to Kettner et al. [18]. The mobile phase consisted of 92.5% (*v*/*v*) 20 mM NaH_2_PO_4_, 10 mM octane-1-sulfonic acid sodium salt (pH adjusted to 2.2 using 4 M H_3_PO_4_), and 7.5% (*v*/*v*) acetonitrile. The injection volume was set to 5 µL. The separation was done at 40 °C at a constant flow rate of 1 mL·min^−1^ for 25 min. The histamine was detected at a wavelength of 210 nm with an ultraviolet detector.

### 2.11. Statistical Analysis

All experiments were performed at least in duplicate and evaluated by determining the standard deviation with Excel. Data are presented as mean values with standard deviation. Enzyme kinetics were evaluated by nonlinear regression using the data-analyzing software Sigmaplot 12.5 (Systat Software GmbH, Erkrath, Germany).

## 3. Results and Discussion

### 3.1. Production of DAO-1 for the Preparation of Tablets

It was estimated in a previous study that high DAO activities of around 50 nkat are required to treat histamine intolerance by oral supplementation [18]. A microbial DAO (DAO-1) was discovered recently and homologously recombinantly produced in the yeast *Y. lipolytica* PO1f in a bioreactor with a working volume of 800 mL [20]. However, this DAO-1 production process is not sufficient enough for the preparation of DAO-1 tablets. Therefore, the bioreactor cultivation was upscaled to 5 L, whereby an optical density (OD_600_) of 53, bio dry mass of 22 g·L^−1^, and wet yeast mass of 93 g·L^−1^ were reached after 56 h of cultivation (Appendix A). From 150 g of wet yeast cells, around 4.8 µkat of total DAO-1 activity was yielded. This equaled a yield of 31 nkat per gram of wet cells, which is around three times higher than in the previous study [20]. The DAO-1 was purified by ammonium sulfate precipitation and hydrophobic interaction chromatography, which yielded a total of 3 µkat with a specific DAO-1 activity of 15 µkat·g_Protein_^−1^. The specific DAO-1 activity was increased three-fold when compared to the recent work (4.7 nkat·g_Protein_^−1^), which is important to provide a highly active DAO-1 extract in the limited space of a tablet.

### 3.2. Stability of DAO-1 in an SIF

The DAO-1 stability under intestinal conditions is of high relevance because DAO-1 was thought to be orally applied to degrade histamine in the human intestine. The intestinal environment was imitated with an SIF, as described in the United States Pharmacopeia [25]. The SIF contained pancreatin, which is an enzyme mixture with different enzyme activities, such as amylases, peptidases, and lipases [26]. The purified DAO-1 was tested in this SIF at 37 °C with and without established food constituents to simulate possible food matrices (Figure 1).

DAO-1 showed poor stability in pure SIF with a half-life period of less than 5 min. The addition of 50 g BSA·L^−1^ and 25 g sucrose·L^−1^ (food matrix SIF 1) improved the half-life period of DAO-1 to around 15 min.

The addition of 25 g BSA·L^−1^, 25 g WPI·L^−1^ and 50 g sucrose·L^−1^ (food matrix SIF 2) further improved the DAO-1 stability to a half-life period of around 30 min. The best stability of DAO-1 in SIF was observed when maintaining the total sucrose and protein concentration at 50 g·L^−1^, while introducing sodium caseinate as a third protein (food matrix SIF 3; BSA, WPI, and sodium caseinate, each at 16.67 g·L^−1^ and 50 g sucrose·L^−1^). The residual DAO-1 activity of 8 ± 0.1% was determined here after 90 min. Thus, the theoretical mean activity over 90 min was 43%.

The results indicated that the stability of DAO-1 in SIF does not only depend on the amount of protein added but also on the type of protein. This can be explained because the pancreatin peptidases’ active sites are mostly occupied by protein substrates other than DAO-1, which reduces the chance of the hydrolysis of DAO-1.

In addition to the stabilizing effect through other proteins by causing a delayed hydrolysis of DAO-1, the sucrose used is also likely to have a stabilizing impact by the reduction in the water activity [27]. In conclusion, the stability of DAO-1 is challenging to assess under true in vivo conditions due to the individually distinct influence of possible constituents present in food matrices. The food matrix SIF 3 was used for all further experiments because it was the most complex and realistic food matrix.

### 3.3. Kinetics of DAO-1 in an SIF

In addition to the stability of DAO-1, its kinetics in SIF are also highly important for histamine degradation. Accordingly, kinetic investigations with DAO-1 were conducted in pancreatin-free and pancreatin-treated food matrix SIF 3 with histamine concentrations ranging from 1.56–50 mM at 37 °C (Figure 2A,B).

The linearization of the Michaelis–Menten kinetics, according to Hanes-Woolf, is shown in the Appendix A. Thus, the pancreatin-free food matrix SIF 3 represented the unhydrolyzed food matrix, whereas the pancreatin-treated food matrix SIF 3 represented its hydrolyzed state in order to investigate the influence of the peptides and free amino acids generated on the DAO-1 kinetics.

In both cases, a substrate inhibition of DAO-1 was recognized for histamine concentrations greater than 12.5 mM. The corresponding *K*_m_ value of DAO-1 for histamine in the pancreatin-free food matrix SIF 3 was 5.7 ± 0.5 mM (R^2^ = 0.99) and thereby slightly higher than the corresponding *K*_m_ value of DAO-1 in PIPES buffer (25 mM, pH 7.2) with a *K*_m_ value of 2.3 ± 0.2 mM [20]. DAO-1 in the pancreatin-treated food matrix SIF 3 showed a corresponding *K*_m_ value of 4.2 ± 0.5 mM (R^2^ = 0.97). Thus, the generation of free amino acids and peptides from the pancreatic action on the food constituents did not affect the kinetics of DAO-1 unfavorably.

The *K*_m_ value of a human DAO was 0.0028 mM [28]. This high affinity toward histamine is beneficial when histamine needs to be regulated at very low concentrations, such as in the peripheral blood [29]. However, in food matrices, where far higher histamine concentrations are found, the histamine degradation capacity of the microbial DAO-1 is adequately higher and therefore sufficient. To put this into perspective, in humans, a plasma histamine concentration of 0.1 mg·L^−1^ was considered to be a concentration that can induce severe anaphylaxis reactions, while foods such as cheese, red wine, and dry-fermented sausages are reported to contain up to 2500, 55, and 358 mg·kg^−1^, respectively [4,9,29].

### 3.4. Tableting of DAO-1

The DAO-1 activity obtained from the disruption and purification of wet yeast cells from around 370 mL of bioreactor volume was used (690 nkat) for the preparation of one DAO-1 tablet. Since this tableting of DAO-1 only serves as a proof of principle at this stage, further research regarding additional excipients to optimize the tableting procedure is necessary. The detailed specifications of the DAO-1 tablet are shown in Table 1.

The SDS-PAGE analysis of the DAO-1 preparation used showed a distinct DAO-1 band at around 75 kDa (Figure 3).

The purified DAO-1 was concentrated almost 90-fold by ammonium sulfate precipitation, whereby no DAO-1 activity was lost. The next step was the freeze-drying of the DAO-1 together with a catalase from *M. lysodeikticus* to improve the stability and activity of DAO-1 under bioconversion conditions by cleaving the disturbing hydrogen peroxide. It was observed in preliminary experiments that the catalase used withstands the freeze-drying process without any considerable enzyme activity loss. Freeze-drying of the purified DAO-1 in sodium phosphate buffer (20 mM, pH 7) with 40 g·L^−1^ sucrose retained the DAO-1 activity completely. The resulting freeze-dried powder containing DAO-1 was compressed in a self-built tablet press to a tablet of 9 × 7 mm.

### 3.5. Quantification of Histamine by RP-HPLC

The quantitative determination of histamine in the food matrix SIF is a challenging task due to the complex sample matrix. The high sucrose and protein contents there, and more importantly, the number of different kinds of peptides and free amino acids generated from the proteolytic digestion disturbs the analysis of histamine by RP-HPLC. A derivatization of histamine with ortho-phthalaldehyde would also lead to the derivatization of various other possible hydrolysis products and is therefore not applicable for this analytical task [30]. Thus, the sample must be purified before the RP-HPLC analysis in order to remove the majority of the foreign compounds.

Large molecules were removed from the crude sample by size exclusion with PD MidiTrap G-25 columns. The histamine in the sample was then bound to a cation exchange material under acidic conditions due to its positive charge [31]. The histamine was separated from other substances by washing out all unbound compounds and eluting it using a pH shift to alkaline conditions. The histamine samples obtained were separated by RP-HPLC without any derivatization (Figure 4).

Histamine standards for the calibration were prepared in the SIF matrix and treated as described above. The calibration showed sufficient linearity within a range of 0.1–2 mM of histamine (R^2^ = > 0.994) (Appendix A). The limit of detection and quantification were 0.5 and 0.65 mM, respectively. The recovery of histamine standards (1.35 mM) was 106.7%, providing sufficient accuracy for the investigation of the histamine bioconversion using DAO-1 tablets.

### 3.6. Histamine Bioconversion Using DAO-1 Tablets

As has already been mentioned, the supplementation of porcine DAO to support the human DAO in the small intestine has been evaluated in several clinical studies, which found that the DAO supplementation reduced histamine-associated physiological symptoms [13,14,15,16,17]. In contrast to the findings of Comas-Basté et al. [32], it was shown in a recent in vitro study that no DAO activity was detectable in a porcine DAO supplement and at least 50 nkat of DAO activity would be required for the degradation of food-relevant histamine amounts of 75 mg [18]. This histamine amount has been used in clinical studies to identify histamine-intolerant humans [13,33]. However, the total DAO activity of 50 nkat was estimated for use in a buffer system. In reality, the lowered DAO activity, kinetics, and stability under intestinal conditions demand the administration of higher DAO-1 activities. Thus, the DAO-1 activity in the tablet was increased to 690 nkat. However, it has to be considered that for each person, depending on the individual intestinal health, different DAO activities might be necessary. DAO-1 was immediately inactivated in a simulated gastric fluid (data not shown). Therefore, the intended DAO-1 tablet should be protected in a gastric acid-resistant capsule shell when administered in vivo.

The prepared DAO-1 tablet reduced the histamine concentration (1.35 mM; 150 mg·L^−1^; 75 mg) applied initially in the food matrix SIF 3 by 29.3 ± 0.8% at 37 °C in 90 min (Figure 5).

This equaled a total degradation of 22 mg of histamine (29.3%). A complete conversion of the same histamine concentration in a previous study was also not possible using DAO-1 in a buffer system, which was attributed to a potential product inhibitory effect [20]. Thereby, the histamine was reduced by around 75% in the previous study [20]. It can be concluded that DAO-1 inactivation in the food matrix SIF 3 was most probably the reason for the weaker histamine reduction compared to the buffer system, since the kinetics differed only slightly between the two systems. Two tablets can be administered instead of one to compensate for the loss of DAO-1 activity through proteolytic digestion. This would most probably result in a more efficient total histamine degradation. However, in order to obtain the DAO-1 activity required, the microbial production should firstly be further improved, investigating alternative expression hosts for the DAO-1 production.

It is important to understand that the benchmark of 75 mg of histamine applied is a theoretical value. This amount was even enough to provoke typical symptoms of histamine intolerance in some healthy individuals [33]. Therefore, the total degradation of 22 mg achieved using one DAO-1 tablet might already be sufficient to help histamine-intolerant humans.

Additionally, vegetal DAO from pea has already been investigated for a possible treatment of histamine-related symptoms under simulated intestinal conditions [34,35,36]. Thereby, the half-life period of pea DAO in an SIF was at around 18 h and thereby higher than the microbial DAO-1 [36]. Indeed, a particular high stability in SIF is beneficial since activity losses do not have to be compensated by the addition of higher DAO activities. However, the application of a microbial DAO might still be the superior approach due to the improved producibility.

Besides the approach of an oral DAO supplementation aiming to degrade histamine in the intestine, another working group focused on the development of a first-in-class histamine-degrading biopharmaceutical for histamine regulation in the peripheral blood [37]. Here, a recombinantly produced human DAO was mutated in its heparin-binding motif to decrease its plasma clearance. However, this application rather targets the treatment of medical conditions such as mast cell activation syndrome, mastocytosis, or anaphylaxis which cause an increased release of endogenous histamine.

To further address the question of how useful the DAO-1 tablets might be, the condition ‘histamine intolerance’, its pathogenetic basis, and its diagnostics must first be entirely understood. Here, evidence-based, double-blind, placebo-controlled, and cross-over in vivo studies are necessary for a deeper understanding of this condition and should be the basis for further investigations. Thus, a DAO-1 tablet with reasonable DAO activity may be useful to help understand the condition of histamine intolerance.

## 4. Conclusions

DAO-1 from *Y. lipolytica* was investigated for its potential in the reduction in histamine under simulated intestinal conditions. Accordingly, the purified DAO-1 was formulated as a sucrose-based tablet containing 690 nkat of DAO-1 activity. The tablet also contained a catalase from *M. lysodeikticus,* ensuring that none of the hydrogen peroxide generated would inactivate DAO-1 during the histamine degradation. It was shown for the first time using a microbial DAO tablet preparation that up to 29.3% (=22 mg) of the applied histamine was degraded under simulated intestinal conditions. This is an impact of histamine degradation that may already be sufficient to circumvent symptoms of histamine intolerance, supporting the endogenous histamine degradation. However, it was also observed that the DAO-catalyzed degradation of histamine in SIF is distinctively interfered by the proteolytic digestion of pancreatin peptidases. This interference varied greatly depending on the type and complexity of the food matrix that was consumed and is thereby challenging to assume for in vivo test systems. Therefore, clinical studies to test the true potential of the DAO-1 tablets in the treatment of histamine intolerance must follow.

## Figures and Tables

**Figure 1 nutrients-14-02621-f001:**
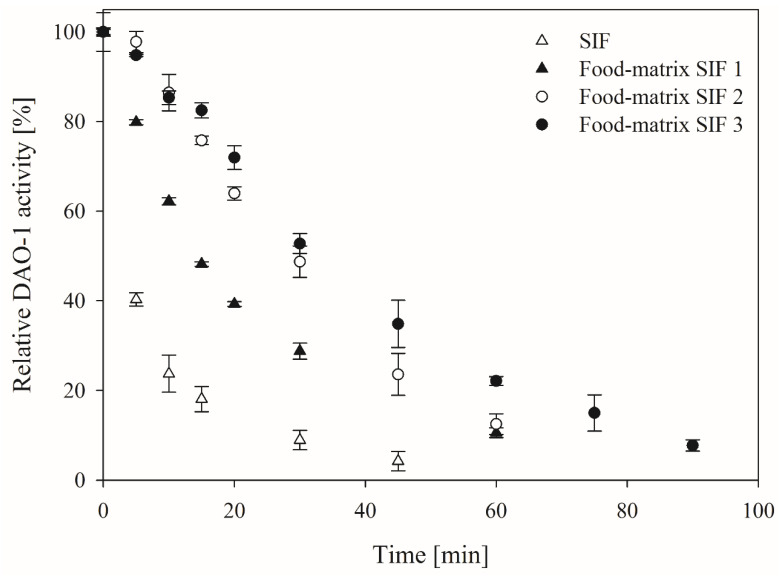
Stability of diamine oxidase (DAO-1) in pure and modified simulated intestinal fluid (SIF). Food matrix SIF 1 contained 50 g bovine serum albumin (BSA)·L^−1^ and 25 g sucrose·L^−1^. Food matrix SIF 2 contained BSA and whey protein isolate (WPI), each at 25 g·L^−1^ and 50 g sucrose·L^−1^. Food matrix SIF 3 contained BSA, WPI, and sodium caseinate, each at 16.67 g·L^−1^ and 50 g sucrose·L^−1^. A total of 100% DAO-1 activity = 1.25 ± 0.15 nkat_Histamine_·mL^−1^.

**Figure 2 nutrients-14-02621-f002:**
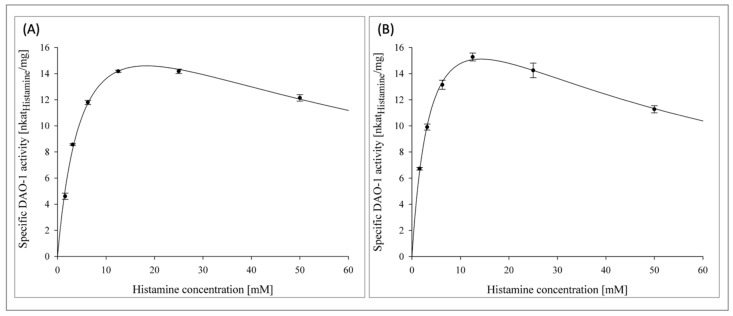
Kinetics of DAO-1 in pancreatin-free (**A**) and pancreatin-treated (**B**) food matrix SIF 3.

**Figure 3 nutrients-14-02621-f003:**
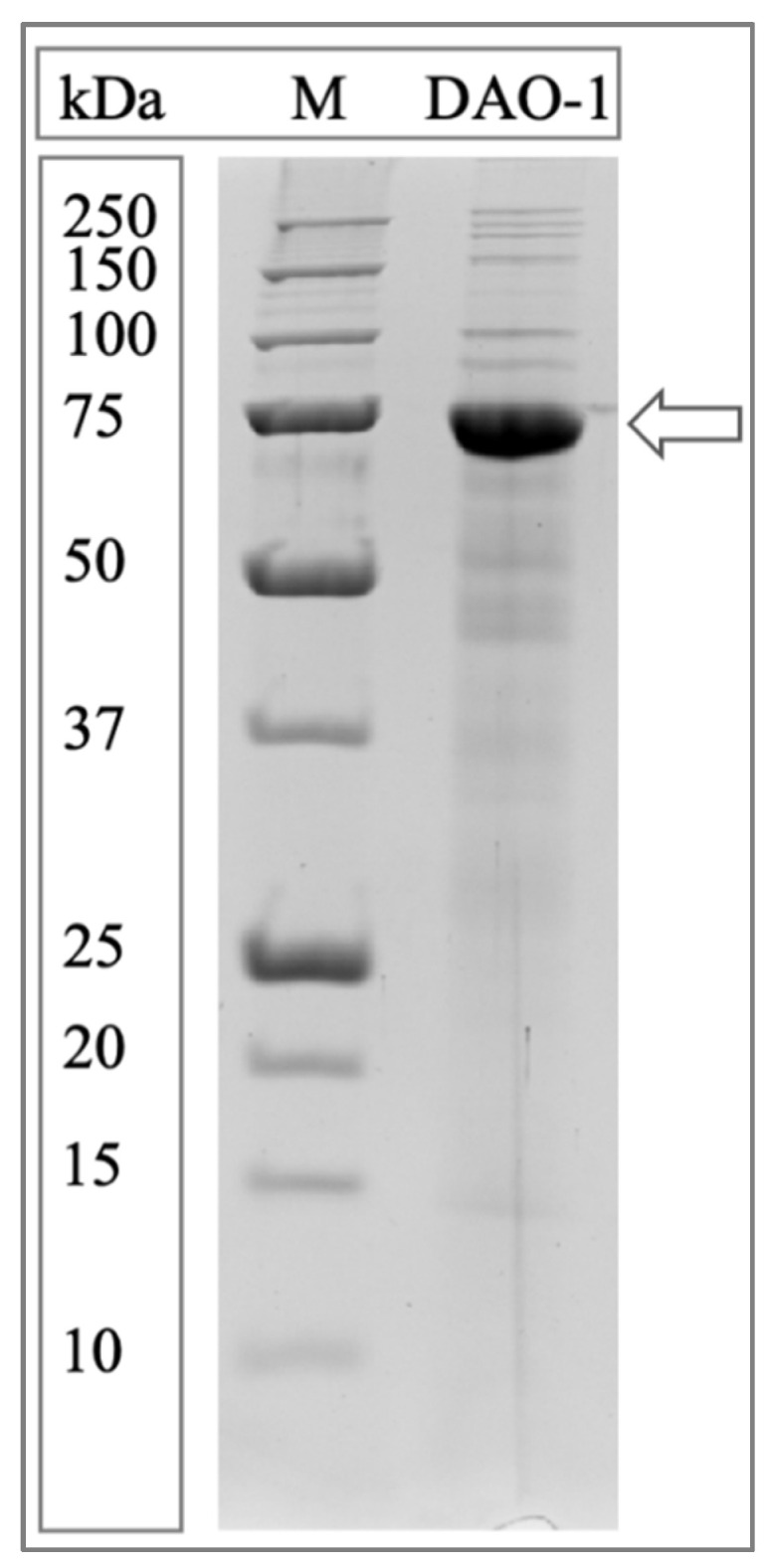
SDS-PAGE analysis of the DAO-1 preparation used for the preparation of DAO-1 tablets. Precision Plus Protein™ unstained protein standard 10–250 kDa (M). Arrow indicates the DAO-1 protein band.

**Figure 4 nutrients-14-02621-f004:**
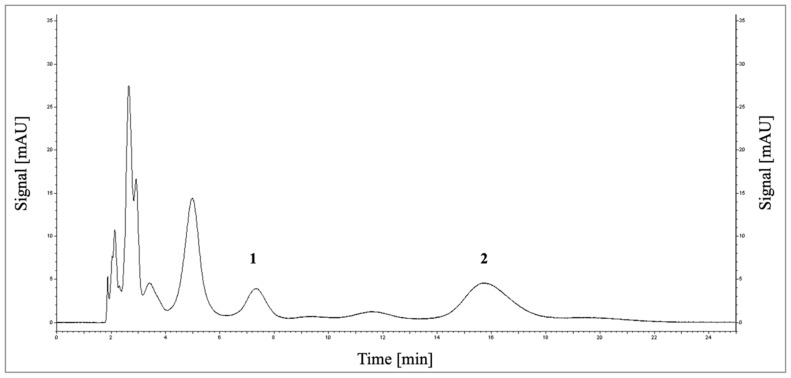
Chromatogram (reversed-phase high-performance liquid chromatography (RP-HPLC)) for the analysis of histamine (**1**) and thiamine (**2**) in the pancreatin-digested food matrix SIF after 90 min.

**Figure 5 nutrients-14-02621-f005:**
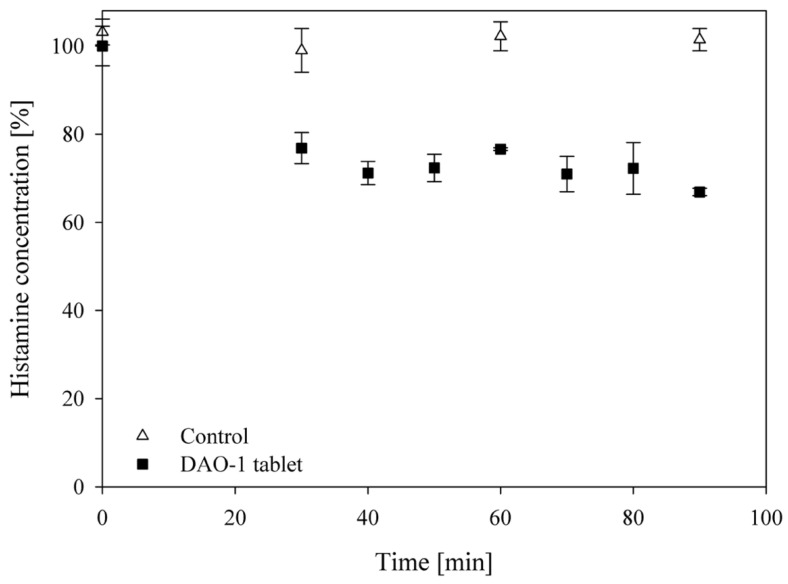
Bioconversion of 75 mg of histamine in food matrix SIF 3 by one DAO-1 tablet (690 nkat DAO-1). For the control, no DAO-1 tablet was applied. Histamine concentrations were determined by RP-HPLC.

**Table 1 nutrients-14-02621-t001:** Specifications of the DAO-1 tablet.

DAO-1 Tablet Specifications (for 1 Tablet)	
DAO-1 activity [nkat]	690
Catalase activity [µkat]	15
Protein [mg]	44
Sucrose [mg]	238
Total weight [mg]	400
Size [mm]	9 × 7 (length × diameter)

## Data Availability

Data supporting the conclusions of this article can be made available by the corresponding author upon reasonable request.

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
