# Peer review of "Toward Oral Supplementation of Diamine Oxidase for the Treatment of Histamine Intolerance"

_nutrients, 2022, doi:10.3390/nu14132621_

Round 1

Reviewer 1 Report

Manuscript Number: 1731933-Toward a diamine oxidase tablet for the treatment of histamine intolerance by oral supplementation

The study presented here is well designed, with interesting results and conclusions. The manuscript is well prepared, it shows in-depth knowledge of the issues raised, both in terms of the research and analytical scope. The research methods are relevant and well described. I believe that the article should be adopted as it stands. Briefly, I find that the manuscript has a merit to be published in "Nutrients" journal once the following recommendations and suggestions are taken into consideration.

Main Comments:

1.         L. 9: "Yarrowia lipolytica" revised to "Yarrowia lipolytica PO1f ".

2.         L. 13: "Yarrowia lipolytica PO1f " revised to "Y. lipolytica ".

3.         L. 25: "The biogenic amine histamine is..." revised to "The histamine, one of biogenic amines, is...".

4.         L. 64: Change to "A new DAO-1 was discovered recently in Yarrowia lipolytica and ...".

5.         P. 5, L.225-230: The author mentioned no derivation was used in HPLC method for histamine analysis. However, histamine compound is very difficult to detect by UV-Vis., because there is not absorption wavelength in UV-Vis. range. Authors were to explain their detail analytical method. For example, which detector was used? which wavelength was set in detector?

6.         L. 382: The sentence "Thereby, the histamine was reduced by around 75%." is unclear. You say "75% reduction" is for your result in this study or result from [20] of reference. In addition, "a total degradation of 22 mg histamine" is for how much % degradation of histamine in this study (L. 379).

7.         L. 402-403: The statement "% degradation of histamine" is better than "histamine amount (22 mg) were degraded".

Author Response

Response to Reviewer 1 comments

Point 1:  L. 9: "Yarrowia lipolytica" revised to "Yarrowia lipolytica PO1f ".

Response 1: We have revised this as suggested (L. 10).

Point 2:  L. 13: "Yarrowia lipolytica PO1f " revised to "Y. lipolytica ".

Response 2: We have revised this as suggested (L. 13).

Point 3:  L. 25: "The biogenic amine histamine is..." revised to "The histamine, one of biogenic amines, is...".

Response 3: We have revised this as suggested (L. 25).

Point 4:  " L. 64: Change to "A new DAO-1 was discovered recently in Yarrowia lipolytica and ...". 

Response 4: We have revised this as suggested (L. 75).

Point 5:  P. 5, L.225-230: The author mentioned no derivation was used in HPLC method for histamine analysis. However, histamine compound is very difficult to detect by UV-Vis., because there is not absorption wavelength in UV-Vis. range. Authors were to explain their detail analytical method. For example, which detector was used? which wavelength was set in detector?

Response 5: Thank you for your advice. We have added the following sentence to the article: “The histamine was detected at a wavelength of 210 nm with an ultraviolet detector.” (L. 244)

Point 6:  L. 382: The sentence "Thereby, the histamine was reduced by around 75%." is unclear. You say "75% reduction" is for your result in this study or result from [20] of reference. In addition, "a total degradation of 22 mg histamine" is for how much % degradation of histamine in this study (L. 379).

Response 6: Thank you for this suggestion. We have modified the sentence to: “Thereby, the histamine was reduced by around 75 % in the previous study [20] (L. 414). Furthermore, we have added the histamine degradation rate in % for this study (29.3 %) (L. 411).

Point 7:  L. 402-403: The statement "% degradation of histamine" is better than "histamine amount (22 mg) were degraded".

Response 7: Thank you for your suggestion. We have modified the sentence to: “It was shown for the first time using a microbial DAO tablet preparation, that up to 29.3 % (= 22 mg) of the applied histamine were degraded under simulated intestinal conditions.” (L. 454)

Reviewer 2 Report

In this manuscript Kettner and co-authors describe the pharmaceutical characteristics of a new diamino-oxidase produced by homologous recombination technology in the yeast Y. lipolytica, including the measurement of its enzymatic activity when formulated as a sucrose-based tablet under simulated intestinal conditions in the presence of a food matrix

The manuscript is well written, the results are interesting, the discussion is elaborated with proper consideration of available literature.

This reviewer does not have the expertise to evaluate the technicalities related to the formulation of the tablet

I would suggest to expand the discussion  with some considerations on histamine intolerance

The same Authors previously (reference 18) quantified the amount of DAO necessary to degrade a food-relevant amount of histamine, established as 75 mg. However, the paper where double blind, placebo controlled, cross-over challenge was used in vivo in healthy volunteers (ref 13) demonstrated that ingestion of 75 mg of liquid histamine failed to reproduce histamine-associated symptoms in many subjects

Moreover, in the reference 18 paper  no measurable amounts of DAO activity was detected by the Authors in commercially available capsules used as food supplementation. These same capsules  demonstrated in a few studies, including ref 13,  clinical activities on histamine intolerance symptoms (ref 14, 15, 16, 17) which, on this basis, cannot be explained as a beneficial consequence of the enzymatic activity of DAO supplementation

With this background in mind, while the rationale for producing pharmaceutical products with meaningful amounts of enzymatically active DAO is warranted, the condition presently indicated as istamine intolerance awaits demonstration of its pathogenetic basis in an evidence-based, in vivo experimental setting, possibly based on double blind, placebo controlled, cross-over histamine challenge. The experimental design should allow to evaluate the possibility that single subjects may have different target organs of istamine reactivity in different occasions. In these conditions, the administration of the new enzymatically active DAO tablets would nicely serve the scope to improve present knowledge on histamine intolerance

Minor points

lines 38-39

"Firstly, single-nucleotide polymorphisms in the DNA sequence of DAO can decrease the expression or kinetics of DAO, respectively, lowering the general histamine-degrading capability [9]."

should be re-phraseed. For instance 

"different single-nucleotide polymorphisms have been associated with lower transcriptional activity of the DAO gene or with reduction of the DAO enzyme functionality, respectively [9]

lines 135-136, line 259

"well-known" should be "established"

lines 238-251

They seems methods, rather than results/discussion

line 247

150/4.8 = 31,25. So "32 nkat" should be approximated to 31

line 306

"blood plasma" should be "peripheral blood"

line 311

"sufficient" should be "adequately higher"

line 307

"was determined to be" should be "was"

line 407

"disturbed" should be "interfered", "disturbance" should be "interference"

Author Response

Response to Reviewer 2 comments

Point 1:  I would suggest to expand the discussion  with some considerations on histamine intolerance

The same Authors previously (reference 18) quantified the amount of DAO necessary to degrade a food-relevant amount of histamine, established as 75 mg. However, the paper where double blind, placebo controlled, cross-over challenge was used in vivo in healthy volunteers (ref 13) demonstrated that ingestion of 75 mg of liquid histamine failed to reproduce histamine-associated symptoms in many subjects. Moreover, in the reference 18 paper  no measurable amounts of DAO activity was detected by the Authors in commercially available capsules used as food supplementation. These same capsules  demonstrated in a few studies, including ref 13,  clinical activities on histamine intolerance symptoms (ref 14, 15, 16, 17) which, on this basis, cannot be explained as a beneficial consequence of the enzymatic activity of DAO supplementation

With this background in mind, while the rationale for producing pharmaceutical products with meaningful amounts of enzymatically active DAO is warranted, the condition presently indicated as istamine intolerance awaits demonstration of its pathogenetic basis in an evidence-based, in vivo experimental setting, possibly based on double blind, placebo controlled, cross-over histamine challenge. The experimental design should allow to evaluate the possibility that single subjects may have different target organs of istamine reactivity in different occasions. In these conditions, the administration of the new enzymatically active DAO tablets would nicely serve the scope to improve present knowledge on histamine intolerance

Response 1: Thank you very much for your remarkable thoughts on this topic. We have included the following paragraph in the article: “To further address the question of how useful the DAO-1 tablets might be, the condition ‘histamine intolerance’, its pathogenetic basis and its diagnostics must first be entirely understood. Here, further evidence-based, double-blind, placebo-controlled, cross-over in vivostudies are necessary for a deeper understanding of this condition and should be the basis for further investigations. Thus, a DAO-1 tablet with reasonable DAO activity could be useful to help understanding the condition histamine intolerance.“ (L. 442)

Point 2:  lines 38-39. "Firstly, single-nucleotide polymorphisms in the DNA sequence of DAO can decrease the expression or kinetics of DAO, respectively, lowering the general histamine-degrading capability [9]." should be re-phraseed. For instance "different single-nucleotide polymorphisms have been associated with lower transcriptional activity of the DAO gene or with reduction of the DAO enzyme functionality, respectively [9]

Response 2: Thank you for your suggestion. We have revised this and changed the sentence to your well-suiting suggested phrasing. (L. 39)

Point 3: lines 135-136, line 259 "well-known" should be "established"

Response 3: We have revised this as suggested. (L. 147; 278)

Point 4:  lines 238-251. They seems methods, rather than results/discussion

Response 4: Thank you for your suggestion. We agree that some parts of this section might not be suitable for the results/discussion part. Therefore, we have modified the sentence “This was followed by cell disruption using a bead mill, whereby 150 g of wet yeast cells were disrupted, yielding around 4.8 µkat of total DAO-1 activity“ to: „From 150 g wet yeast cells, around 4.8 µkat of total DAO-1 activity were yielded.“ (L. 260)

Point 5: line 247, 150/4.8 = 31,25. So "32 nkat" should be approximated to 31

Response 5: We have revised this as suggested. (L. 261)

Point 6: line 306, "blood plasma" should be "peripheral blood"

Response 6: We have revised this as suggested. (L. 329)

Point 7: line 311, "sufficient" should be "adequately higher"

Response 7: Thank you for your suggestion. We have modified the sentence to: “However, in food matrices, where far higher histamine concentrations are found, the histamine degradation capacity of the microbial DAO-1 is adequately higher and therefore sufficient.“ (Line 330)

Point 8: line 307, "was determined to be" should be "was"

Response 8: We have revised this as suggested. (L. 327)

Point 9: line 407, "disturbed" should be "interfered", "disturbance" should be "interference"

Response 9: We have revised this as suggested. (L. 459)

Reviewer 3 Report

1. Presenting the results of their research, the authors do not analyze the existing developments in this field of medicine and pharmacy. Preparations, biologically active additives of this type already exist and are used, but the authors do not provide any literature data on this.

2. The title of the article indicates the development of a diamine oxidase tablets, but the manuscript contains a lot of material on the development of the pharmaceutical substance itself, the search for conditions for its stability, etc. In our opinion, the title of the manuscript needs to be corrected.

3. When developing model systems for their research, the authors do not take into account the conditions of intestinal pathology, the possibility of hydrolysis of the enzymes in the stomach, etc. The data obtained are relevant only for the conditions of DAO-1 transformations in a healthy intestine, without taking into account the transport of the drug through other parts of the gastrointestinal tract.

4. The manuscript does not contain data on excipients that are necessary for the preparation of tablets, i.e. Table 1 shows the composition of not tablets, but only the active pharmaceutical substance for their preparation without the necessary excipients.

5. The authors' studies are fragmentary, and the results do not allow us to state the fact of the creation of a pharmaceutical substance, and even more so tablets based on it. So far, these are preliminary studies that need to be continued both in vitro and in vivo in order to obtain more meaningful results. In addition, it is necessary to use comparators in these studies that already exist and are used in such a pathology.

Author Response

Response to Reviewer 3 comments

Point 1: Presenting the results of their research, the authors do not analyze the existing developments in this field of medicine and pharmacy. Preparations, biologically active additives of this type already exist and are used, but the authors do not provide any literature data on this.

Response 1: Thank you very much for your helpful comment. We have included further literature about the use of vegetal DAO and a recombinantly produced human DAO for the treatment of certain histamine-related medical conditions. We have added the following paragraph to the manuscript: “Also, a vegetal DAO from pea has already been investigated for a possible treatment of histamine-related symptoms under simulated intestinal conditions [34,35,36]. Thereby, the half-life period of pea DAO in a SIF was at around 18 h and thereby higher than the microbial DAO-1 [36]. Indeed, a particular high stability in SIF is beneficial since activity losses do not have to be compensated by the addition of higher DAO activities. However, the application of a microbial DAO might still be the superior approach due to the improved producibility. Besides the approach of an oral DAO supplementation aiming to degrade histamine in the intestine, another working group focused on the development of a first-in-class histamine-degrading biopharmaceutical for the histamine regulation in the peripheral blood [37]. Here, a recombinantly produced human DAO was mutated in its heparin-binding motif to decrease its plasma clearance. However, this application rather targets the treatment of medical conditions like mast cell activation syndrome, mastocytosis or anaphylaxis that cause an increased release of endogenous histamine.” (L. 428)

Point 2: The title of the article indicates the development of a diamine oxidase tablets, but the manuscript contains a lot of material on the development of the pharmaceutical substance itself, the search for conditions for its stability, etc. In our opinion, the title of the manuscript needs to be corrected.

Response 2: Thank you for your constructive advice. We agree that the title should be changed in order to take the focus away from the tablet itself. Therefore, we suggest the title: “Toward oral supplementation of diamine oxidase for the treatment of histamine intolerance”.

Point 3: When developing model systems for their research, the authors do not take into account the conditions of intestinal pathology, the possibility of hydrolysis of the enzymes in the stomach, etc. The data obtained are relevant only for the conditions of DAO-1 transformations in a healthy intestine, without taking into account the transport of the drug through other parts of the gastrointestinal tract.

Response 3: Thank you for your comment. We have included a section describing the stability of DAO-1 in a simulated gastric fluid and the necessity to provide a gastric-acid resistant capsule shell to transport the DAO-1 tablet to the intestine. “However, it has to be considered that for each person, depending on the individual intestinal health, different DAO activities might be necessary. The DAO-1 was immediately inactivated in a simulated gastric fluid (data not shown). Therefore, the intended DAO-1 tablet should be protected in a gastric-acid resistant capsule shell when administered in vivo.“ (L. 399)

Point 4: The manuscript does not contain data on excipients that are necessary for the preparation of tablets, i.e. Table 1 shows the composition of not tablets, but only the active pharmaceutical substance for their preparation without the necessary excipients.

Response 4: Thank you for your comment. Indeed, the DAO-1 tablet at this stage can be seen as a proof of principle and further research is required to establish a better tableting procedure. Therefore, we have included the following sentence in the manuscript: “Since this tableting of DAO-1 only serves as a proof of principle at this stage, further research regarding additional excipients to optimize the tableting procedure is necessary.“ (L. 343)

Point 5: The authors' studies are fragmentary, and the results do not allow us to state the fact of the creation of a pharmaceutical substance, and even more so tablets based on it. So far, these are preliminary studies that need to be continued both in vitro and in vivo in order to obtain more meaningful results. In addition, it is necessary to use comparators in these studies that already exist and are used in such a pathology.

Response 5: Thank you for your comment. We agree that the results shown in the manuscript are a first proof of principle and that further studies both in vitro and especially in vivo should follow. We have added the following paragraph: “To further address the question of how useful the DAO-1 tablets might be, the condition ‘histamine intolerance’, its pathogenetic basis and its diagnostics should first be entirely understood. Here, evidence-based, double-blind, placebo-controlled, cross-over in vivo studies are necessary for a deeper understanding of this condition and should be the basis for further investigations. Thus, a DAO-1 tablet with reasonable DAO activity could be useful to help understanding the condition histamine intolerance.” (L. 442)

Round 2

Reviewer 3 Report

The corrected version has been reviewed by me.

No questions.